# Oral Anticoagulation in Patients with Chronic Kidney Disease and Non-Valvular Atrial Fibrillation: The FAERC Study

**DOI:** 10.3390/healthcare10122566

**Published:** 2022-12-17

**Authors:** Marco Montomoli, Lourdes Roca, Mariana Rivera, Raul Fernandez-Prado, Beatriz Redondo, Rosa Camacho, Cayetana Moyano, Saul Pampa, Angela Gonzalez, Juan Casas, Maria Kislikova, Ana Sanchez Horrillo, Alicia Cabrera Cárdena, Borja Quiroga, Cristina Rabasco, Sara Piqueras, Andrea Suso, Javier Reque, Juan Villa, Raquel Ojeda, David Arroyo

**Affiliations:** 1Nephrology Department, Hospital Clínico Universitario de Valencia, Av. Blasco Ibañez 17, 46010 Valencia, Spain; 2Nephrology Department, Hospital Universitario de La Plana, 12540 Villarreal, Spain; 3Nephrology Department, Hospital Universitario Virgen de la Macarena, 41009 Sevilla, Spain; 4Nephrology Department, Hospital Universitario Fundación Jiménez Díaz, 28040 Madrid, Spain; 5Nephrology Department, Hospital Universitario de Cruces, 48903 Bilbao, Spain; 6Nephrology Department, Hospital Universitario Severo Ochoa, 28914 Leganes, Spain; 7Nephrology Department, Hospital Universitario Reina Sofía, 14004 Córdoba, Spain; 8Nephrology Department, Hospital Universitario Rey Juan Carlos, 28933 Móstoles, Spain; 9Nephrology Department, Hospital Universitario Gregorio Marañón, 28007 Madrid, Spain; 10Nephrology Department, Hospital Comarcal Francesc de Borja, 46702 Gandía, Spain; 11Nephrology Department, Hospital Universitario Marqués de Valdecilla, 39008 Santander, Spain; 12Nephrology Department, Hospital Universitario de La Princesa, 28006 Madrid, Spain; 13Nephrology Department, Complejo Hospitalario Universitario de Albacete, 02006 Albacete, Spain; 14Nephrology Department, Hospital General Universitario de Castellón, 12004 Castelló, Spain; 15Nephrology Department, Hospital Universitario de Badajoz, 06080 Badajoz, Spain

**Keywords:** chronic kidney disease, atrial fibrillation, prevalence, direct-action anticoagulants, coumarins, mortality, ischemic stroke, haemorrhage

## Abstract

Atrial fibrillation (AF) is the most common arrhythmia in patients with chronic kidney disease (CKD), and its presence is associated with a higher risk of stroke and mortality. Material and Methods: The FAERC study performed a retrospective multicentre analysis of historical cohorts in which data were collected from arrhythmia diagnosis onwards. Results: We analysed a Spanish cohort of 4749 patients with CKD (mean eGFR 33.9 mL/min) followed up in the nephrology clinic, observing a 12.2% prevalence of non-valvular AF. In total, 98.6% of these patients were receiving anticoagulant treatment, mainly with coumarins (79.7%). Using direct-acting oral anticoagulants (DOACs) was associated with fewer cerebrovascular events than using acenocoumarol, but in contrast with other studies, we could not corroborate the association of risk of bleeding, coronary events, or death with a type of anticoagulant prescribed. Conclusions: Atrial fibrillation is highly prevalent in renal patients. Direct-acting anticoagulants seem to be associated with fewer ischemic-embolic complications, with no differences in bleeding, coronary events, or mortality rates.

## 1. Introduction

Atrial fibrillation (AF) is the most common arrhythmia in patients with chronic kidney disease (CKD) [1]. CKD was included as a risk factor for developing AF in the 2016 European Society of Cardiology (ESC) guidelines [2]. Its prevalence increases proportionally according to the decrease in estimated glomerular filtration rate (eGFR) and the presence of albuminuria [3], to the extent that its prevalence in patients with CKD in advanced stages (ACKD) is threefold higher than in the general population [1]. Finally, patients with CKD and AF have a higher risk of stroke and mortality than CKD patients without AF [4,5].

Vitamin K antagonist anticoagulants (VKAs—the coumarins warfarin and acenocoumarol) have been in use for over 50 years, and the other more recently appearing drugs which inhibit factor Xa (rivaroxaban, apixaban, edoxaban) and thrombin (dabigatran) are known as direct-acting oral anticoagulants (DOACs).

Despite the increase in the arsenal of anticoagulant drugs, the treatment of choice for patients with ACKD has not yet been established. There is a concern about increased vascular calcification and calciphylaxis with VKAs, given that they reduce the function of vitamin K–dependent vascular calcification inhibitors, and about the possibility of AKI secondary to glomerular haemorrhage due to VKAs, and to a lesser extent to DOACs, in patients in whom there is no other identifiable aetiology of AKI [6].

Furthermore, the pharmacokinetics and pharmacodynamics of most anticoagulant drugs are significantly affected by renal dysfunction, and they have significant drug–drug interactions that are especially important given the polypharmacy that is so prevalent in patients with CKD [6].

However, studies published to date show a lower rate of stroke/systemic embolism with DOACs compared to coumarins, as well as a lower risk of renal events (acute renal failure and progression of CKD) and possibly of anticoagulant-associated nephropathy (AAN) in patients with mild-moderate CKD [7,8].

There is a notable lack of published clinical trials in patients with CKD stages 4, 5 and 5D in treatment with DOACs. We, therefore, conducted the Atrial Fibrillation in Chronic Kidney Disease (FAERC) study, in which we measured the prevalence of AF in CKD patients without renal replacement therapy under follow-up in the nephrology clinics of different hospital centres in Spain, and evaluated the anticoagulant treatment used in real clinical practice and its clinical implications, focusing on prevention of ischemic-embolic and haemorrhagic risk.

An appreciation of the risks associated with anticoagulation and an awareness of the advantages and disadvantages of currently available anticoagulants are key factors in the decision-making process.

## 2. Material and Methods

### 2.1. Study Design

The FAERC project was designed as a retrospective, multicentre, observational study of historical cohorts in which data was collected from arrhythmia diagnosis onwards (until the last visit consultation on 30 June 2020). Fifteen hospitals from 10 Spanish provinces participated. The inclusion criteria were age 18 years old or above; CKD of any stage according to the definition of the Kidney Disease: Improving Global Outcomes (KDIGO) guidelines [9]; diagnosis of paroxysmal or persistent nonvalvular AF, according to coding in the patient’s clinical history; and at least one follow-up visit in the nephrology clinic between January 2019 and June 2020.

According to the Spanish consensus document for the detection and management of chronic kidney disease, patients with eGFR < 30 mL/min/m^2^ or, among others, with higher eGFR but rapid progression (creatinine albumin ratio > 300 mg/g or decrease in eGFR > 5 mL/min/year) are usually referred to the nephrology clinic [10].

Patients on renal replacement therapy, including kidney transplant, with valvular disease defined as moderate to severe mitral or aortic regurgitation, with stenosis, or prosthetic valve carriers were excluded from the study. Patients were registered consecutively throughout the study period, and the total number of patients with CKD who attended each hospital consultation in that same period was collected to calculate the prevalence of AF. Clinical variables (age, sex, comorbidities, CKD aetiology, history of cardiovascular events (CVD)) and analytical variables (glomerular filtration rate estimated by CKD-EPI, albuminuria, expressed by the albumin/creatinine ratio in urine and haematuria, defined as more than 5 RBCs per field in sediment analysis) were collected at the time of AF diagnosis. The different anticoagulant treatments prescribed for AF during follow-up were recorded, as were ischemic and haemorrhagic complications. CVDs were defined as unstable angina, myocardial infarction or coronary revascularization, ischemic stroke, or symptomatic peripheral artery disease requiring amputation. Serious haemorrhagic complications were defined as a haemoglobin concentration drop of 2 g/dL, transfusion requirements, cerebral haemorrhages or those requiring surgical treatment, and death due to bleeding. Changes in renal function were also recorded at 1 year after AF diagnosis and at the end of follow-up; patients with a drop in eGFR above the median were considered “rapid progression”. Due to its observational nature, the study did not require funding.

The study was approved, with the code 21/20, by the Medical Research Ethics Committee of Dr Peset University Hospital in Valencia on 30 March 2020. The same committee approved the exemption from requesting informed consent from patients due to the study’s retrospective nature. The main authors of the study conducted the statistical analysis. The first draft of the manuscript was written by the first and last authors and was revised by the co-authors. The decision to submit the manuscript for publication was made jointly by all of the authors. The first and last authors vouched for the data’s completeness and accuracy and for the protocol’s fidelity to the trial.

### 2.2. Statistical Analysis

Statistical analysis was carried out with IBM SPSS Statistics version 20.0 (Chicago, IL, USA). Qualitative variables were described by their absolute and relative frequency, and quantitative variables by mean and standard deviation if they had a normal distribution, or median and interquartile range otherwise. The normality of the sample was checked using the Kolmogorov–Smirnov test. Qualitative variables were associated with the χ^2^ test, and Student’s *t*-test and Mann–Whitney U test were used for mean comparisons. The Mantel–Cox test (or log-rank test) studied the association of factors with time-dependent variables in the different groups. A bilateral *p* of less than 0.05 was considered significant.

## 3. Results

### 3.1. Description of Population Characteristics

Data from 578 patients with CKD and nonvalvular AF were collected out of a total of 4749 CKD patients treated during the period, revealing a mean AF prevalence of 12.2%. However, a wide variation between centres was observed (5.9–20.0%). The cohort included 364 (63.0%) men with a median age of 77.6 ± 8.8 years at AF diagnosis. Clinical and analytical variables recorded are shown in Table 1. In summary, the median estimated GFR was 49.2 ± 20.1 mL/min. The eGFR in the blood samples was calculated by CKD-EPI according to the CKD KDIGO guidelines in all visit controls [9].

Vascular pathology was the most frequent aetiology of kidney disease (30.1%).

Median albuminuria was 150.0 (30.5–500.0) mg/g, and 20.4% of patients had haematuria. Among comorbidities, arterial hypertension (93.6%), dyslipidaemia (72.6%), diabetes mellitus (47.6%) and heart failure (45.3%) were the most frequent.

### 3.2. Anticoagulant Treatment

Some type of anticoagulation was indicated in 98.6% of patients, especially coumarins (79.7%). Notably, only 0.3% of anticoagulant treatments were prescribed by nephrologists, whereas cardiologists were the most common prescribers (58.7%).

Prescribed treatments and changes during follow-up are summarized in Table 2.

We did not find any difference in prescription when we analysed the prevalence of kind of anticoagulation, at the start of the study, according to eGFR.

During follow-up, the number of patients treated with DOACs increased significantly (34.6 vs. 15.9%), at the expense of a decrease in coumarins to 58.8%.

### 3.3. Complications during Follow-Up

Median follow-up from AF diagnosis to the most recent consultation was 61.0 months (33.0–101). During this time, 34 patients (5.9%) had an ischemic or embolic event despite anticoagulant treatment. Of these, 30 (88.2%) were cerebrovascular accidents (CVA), three (8.8%) systemic embolisms, and one case (3.0%) pulmonary thromboembolism. When analysing the factors associated with a higher risk of ischemic-embolic events in the Cox regression model (Figure 1), we found a statistically significant association with a history of previous stroke (27.2 vs. 0.6%, HR 56.4, 95% CI 16.2–191.2) and use of coumarins compared to DOACs (6.5 vs. 1.1%, HR 9.2, 95% CI 1.2–71.4). Bleeding complications were frequent, seen in 163 patients (28.2%), of which 107 (18.5%) met the criteria for major bleeding. In 35 patients (6.1%), anticoagulant treatment was discontinued. The type of anticoagulant was not associated with an increased risk of bleeding. Fifty-eight patients (10%) died during follow-up, the most frequent causes being infectious (30.4%), cardiovascular (25.0%) and neoplastic (12.5%). The predictors of mortality were diabetes mellitus (HR 2.7, 95% CI 1.1–3.9), bleeding episodes (HR 2.6, 95% CI 1.4–4.9) and coronary ischemic events (HR 2.4, 95% CI 1.1–5.2%). The type of anticoagulant was not associated with the risk of mortality.

We did not find any difference when we analysed the previous data taking into account the censored data.

### 3.4. Changes in Kidney Function

Comparing the two treatments, patients initially treated with DOACs were younger (75.9 ± 8.9 vs. 78.0 ± 8.8 years, *p* = 0.040) and had worse renal function (eGFR 44.6 ± 20.1 vs. 50.5 ± 16.2 mL/min, *p*= 0.015) than those treated with coumarins (Table 1). Examining renal function decline, we found that 46.5% reached advanced stages of CKD (ACKD). The drop in eGFR in the first year of treatment was greater in the DOAC than coumarin group: −5.1 (−1.0, −10.8) vs. −3.0 mL/min (−0.6, −4.5); the DOAC treatment is a predictor of progression above the median in the binary logistic regression model (HR 2.3, 95% CI 1.2–4.6). However, we did not observe a relationship between anticoagulant type and rate of renal progression throughout follow-up (Figure 2).

### 3.5. Sub Analysis

We analysed a patient subset without changes in anticoagulant treatment throughout follow-up (n = 312). As in the total study population, comparing patients treated with DOACs with those treated with VKAs, no significant difference was observed in survival analysis (log-rank 95% CI, *p* = 0.939). There were also no statistically significant differences in the number of ischemic (log-rank *p* = 0.255) or haemorrhagic events (log-rank *p* = 0.279). In contrast, an ANOVA analysis of repeated eGFR measurements throughout follow-up revealed significant differences in dynamics (GreenHouse–Geisser, 95% CI *p* = 0.002) between the start point and at 12 months: the two groups showed a similar drop (*p* = 0.133) with a significant falling slope (*p* < 0.001). Even between the year of follow-up initiation and the end of the study, patients in treatment with DOACs maintained a slight non-significant fall in eGFR (*p* = 0.124), while it fell significantly in those treated with VKAs (*p* < 0.001).

## 4. Discussion

The results of our study in the CKD population followed up in the nephrology consultation show that the mean prevalence of AF is 12.2%. Other studies in the renal setting show a higher prevalence: an American study, the Chronic Renal Insufficiency Cohort (CRIC), reported a prevalence of 18% in patients older than 65 years. The lower prevalence in our study was despite the more reduced eGFR ((33.9 +/−15.7) vs. 43.6 (+/−13.0) mL/min)) and older average age ((58.55 (+/−10) vs. 77.6 (+/−8.9) years)) in our cohort [1]. In Spain, a recent publication analysing data from the BIG-PAC electronic registry found an AF prevalence of between 16.9 and 17.5% in patients with CKD stage 3–5 [11]. The main limitation of the registry is that CKD diagnosis was made using a single analytical eGFR determination, whereas CKD is defined by deterioration in renal function persisting for at least 3 months [9].

One explanation for our lower results could be that referral patients who start follow-up in the nephrology clinic are not representative of all patients with CKD [10]. Furthermore, our study had wide between-centre variation and a predominance of vascular aetiology, which probably reflects the existence of monographic follow-up consultations for patients with different types of CKD, and perhaps selection bias in patients with lower risk of FA. As suggested by the main AF management guidelines [2], which underline the clear benefits of anticoagulating patients with eGFR > 30 mL/min, almost all patients (98.6%) were prescribed anticoagulant treatment. The initial treatment chosen was predominantly coumarins (79.7%). However, the percentage of patients treated with DOACs doubled along with follow-up (from 15.9% to 34.6%). The final figure was lower than was observed in other studies [12] and should prompt review, taking into account that the latest European and American guidelines suggest recommending this type of drug over coumarins for AF treatment with eGFR > 30 mL/min [13,14]. One obvious cause underlying low DOAC use is the minimal involvement of nephrologists in the therapeutic management of AF anticoagulation (treatment prescriptions by nephrology clinics totalled only 0.9%). In addition, these drugs require careful dose adjustment according to renal function, which can be an obstacle to prescription by other specialists. One in every three patients changed the type of anticoagulant medication during follow-up, mainly due to the difficulty of staying within the international normalized ratio (INR) therapeutic range (44.7%) and the appearance of adverse effects (37.7%). The difficulty of maintaining INR figures in the therapeutic range is highly prevalent in patients with CKD, as deduced from the SCREAM observational study, where 30–40% of patients with eGFR >30 mL/min did not achieve optimal results (defined as the time in therapeutic range > 75%) [15].

DOACs have proven beneficial in reducing the number of cardioembolic and haemorrhagic events in the general population, in both clinical trials and real-life studies [16,17,18,19,20,21]. The evidence in patients with CKD was obtained exclusively from sub-analyses of the cited studies, in which a clear benefit was shown up to a clearance of 30 mL/min, but with contradictory evidence at lower levels [22,23,24].

Although not completely conclusive, the results of our study seem to go in this direction. Patients treated with DOACs had a lower number of CVDs than those treated with acenocoumarol, and aside from eGFR, these results were not confirmed in the patient group with unmodified treatment throughout follow-up.

However, our results go in the same direction as previously published studies in which the use of acenocoumarol and warfarin (aVK) was associated with an increase in systemic calcification, including in the coronary and peripheral vasculature. This increase in vascular calcification is due to the inhibition of the enzyme matrix Gla protein (MGP). MGP is a vitamin K-dependent protein that normally prevents systemic calcification by removing calcium phosphate from tissues. aVK-induced systemic calcification would be related to an increase in vascular events (cardiac and peripheral) [25].

The study also highlights the high risk of bleeding episodes in anticoagulated CKD patients (163 events, equivalent to 5.6 events/100 patients/year).

These data resemble those of a large Swedish registry (SWEDEHEART) collected before DOACs were available (2003–2010) (6.8 events/100 patients/year in patients with CKD stage 3 and 9.1 in CKD stage 4) [26].

In our study, the different risks of bleeding, coronary events or death according to the type of anticoagulant prescribed were not corroborated, although the number of these coronary events was insufficient to allow adequate interpretation.

We observed rapid progression of CKD in the patients included in our study (median drop in GFR was −3.6 mL/min/year (−0.6–−4.8)), perhaps owing to the high cardiovascular risk of this cohort. The decline in real function is common in patients with CKD, and AF and is correlated with increased risk of cardiovascular and haemorrhagic events and mortality, as well as inadequate anticoagulant dose prescription (under- or overdosing), with the risks that this implies [12,27,28].

In our study, patients with DOACs had a greater drop in eGFR in the first year and during follow-up. These data contrast with those published in post hoc analyses of the ROCKET (rivaroxaban) and RE-LY (dabigatran) trials, which found evidence of a possible protective effect of DOACs on CKD progression compared to warfarin [29,30].

These differences could be due to different baseline characteristics of patients: those who started treatment with DOACs were younger and had a lower eGFR. In multivariate analysis, only classic cardiovascular risk factors, but not anticoagulant type, were independent predictors of accelerated progression of renal dysfunction. In addition, analysing patients who had not changed treatment throughout follow-up, we observed that the progression of CKD was slower in patients treated with DOACs.

The FAERC study provides an updated snapshot of AF prevalence and management in CKD patients without renal replacement therapy. Its main strengths are the large sample size and multicentre nature, but the study also has important limitations. Among these, the retrospective and multicentre nature of the study limits the scope of the database, which, therefore, lacks several data of great potential interest for studying these cohorts, such as the CHADS-VASc and HAS-BLED scores, anticoagulant doses and the exact time in treatment with each type (time in therapeutic range). In addition, since the time of changing the anticoagulant was not available, an analysis of complications was made during follow-up by intention to treat, based on the first anticoagulant prescribed. Finally, another drawback is the low number of events during follow-up, which limits the number of statistical inferences possible. Nonetheless, in our view, this study makes a valuable contribution, responding to a still-unmet need for evidence in clinical practice to guide therapeutic decisions, and to the absence of randomized clinical trials in patients with ACKD.

## 5. Conclusions

Atrial fibrillation is highly prevalent in kidney patients. Direct-acting anticoagulants seem to be associated with fewer ischemic-embolic complications, with no differences in rates of bleeding, coronary events, or mortality. Although the use of new anticoagulants has increased, coumarins continue to prevail. The involvement of nephrologists in patient anticoagulation management is, therefore, essential.

## Figures and Tables

**Figure 1 healthcare-10-02566-f001:**
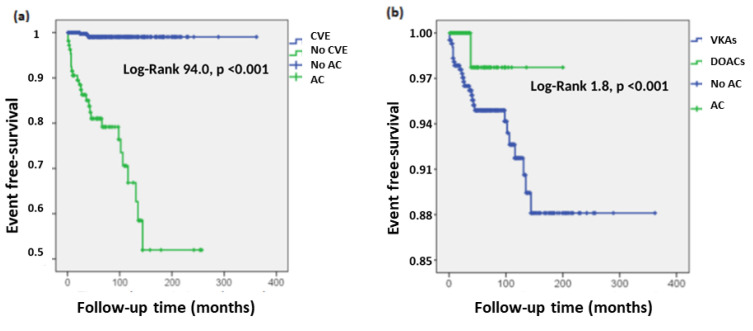
Effect of (**a**) previous cerebrovascular events and (**b**) the use of coumarins compared to DOACS were associated with a higher risk of ischemic and embolic events in the Cox regression model. Abbreviations: DOACS: direct-acting oral anticoagulants. CVE: cerebrovascular event. AC: adjustment for censoring. VKAS: vitamin K antagonist anticoagulants.

**Figure 2 healthcare-10-02566-f002:**
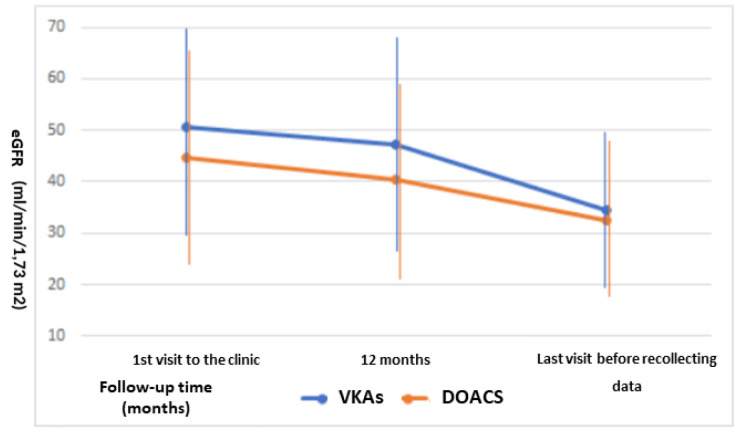
Evolution of renal function depending on the type of anticoagulant: The drop in eGFR in the first year of treatment was greater in the DOAC than VKAs group. However, we did not observe a relationship between anticoagulant type and rate of renal progression throughout follow-up. Abbreviations: eGFR: estimated glomerular filtration rate. DOACs: direct-acting oral anticoagulants. AVK: vitamin K antagonist anticoagulants.

**Table 1 healthcare-10-02566-t001:** Baseline characteristics of the patients included.

	TOTAL N = 578	Acenocoumarol N = 460	DOACs N = 92	*p* Value
Age at the recruitment (years)	77.6 ± 8.8	78.0 ± 8.8	75.9 ± 8.9	0.040
Female sex (n, %)	214 (37.0%)	172 (37.4%)	34 (37.0%)	NS
eGFR (mL/min)				
- Basal	49.2 ± 20.5	50.6 ± 20.3	44.7 ± 20.7	0.015
- One year	45.8 ± 19.2	47.2 ± 19.3	40.3 ± 18.9	0.004
Last visit before recollecting data	33.9 ± 15.7	34.4 ± 16.2	32.5 ± 14.1	NS
ACR (mg/g)				
- Basal	62.0 (10.0–288.5)	71.5 (10.0–250.0)	40.6 (10.0–250.0)	NS
- One year	75.4 (12.0–269.9)	72.7 (12.0–275.4)	50.0 (9.5–157.0)	NS
- Final	150 (30.5–500.0)	150.0 (30.0–500.0)	104.5 (18.0–583.8)	NS
Haematuria (n, %)	118 (20.4%)	103 (22.4%)	11 (12.0%)	NS
Aetiology (n, %)				NS
- Vascular	173 (30.1)	146 (31.8)	19 (20.9)	
- Multifactorial	130 (22.6)	97 (21.1)	27 (29.7)	
- Diabetes	83 (14.4)	68 (14.8)	12 (13.2)	
- TIN	36 (6.3)	31 (6.8)	4 (4.4)	
- Glomerular	32 (5.6)	24 (5.2)	5 (5.5)	
- Hereditary	12 (2.1)	6 (1.3)	6 (6.6)	
- Others	68 (11.9)	36 (7.8)	6 (6.6)	
- Not affiliated	67 (11.7)	51 (11.1)	12 (13.2)	
Comorbidities (n, %)				NS
- Hypertension	540 (93.6)	429 (93.3%)	86 (94.5%)	
- Dyslipidaemia	418 (72.6)	341 (74.5%)	64 (69.6%)	
- Diabetes	275 (47.6)	218 (47.4%)	47 (51.1%)	
- Heart Failure	261 (45.3)	209 (45.6%)	41 (44.6%)	
- Ischemic cardiopathy	156 (27.0)	117 (25.4%)	32 (34.8%)	
- Cerebrovascular event	114 (19.7)	88 (19.1%)	20 (21.7%)	
- Peripherical arteriopathy	76 (13.4)	62 (13.6%)	10 (11.4%)	

Abbreviations: DOACs, direct-acting oral anticoagulants; ACR, albuminuria—creatininuria ratio; eGFR, estimated glomerular filtration rate with CKD-EPI formula; NS, no significance; TIN, tubulointerstitial nephropathy.

**Table 2 healthcare-10-02566-t002:** Anticoagulants prescribed for the treatment of atrial fibrillation.

- Clinician (n, %):	
- **Cardiologist**	339 (58.7)
**- General Practitioner**	66 (11.4)
- **Internal Medicine**	42 (7.3)
- **Nephrologist**	2 (0.3)
- First treatment (n, %)	
- Acenocoumarol	460 (79.7)
- DOACs	92 (15.9)
- LWMH	17 (2.9)
- None	8 (1.4)
- Changes in the treatment (n, %)	181 (31.3)
- Poor control or difficulty with INR	76 (44.7)
- Haemorrhage events	38 (22.4)
- Ischemic events	18 (10.6)
- Other complications	8 (4.7)
- Most recent treatment (n, %)	
- Acenocoumarol	340 (58.8)
- DOACs	200 (34.6)
- LWMH	19 (3.3)
- None	19 (3.3)

Abbreviations: DOACs, direct-acting oral anticoagulants; LWMH, low-weight molecular heparin.

## Data Availability

Data set is available in: https://drive.google.com/file/d/1wtH7lII7D8DY_4R9C-3JnIgLbQKJ2qB_/view?usp=share_link.

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
