# Peer review of "Oral Anticoagulation in Patients with Chronic Kidney Disease and Non-Valvular Atrial Fibrillation: The FAERC Study"

_healthcare, 2022, doi:10.3390/healthcare10122566_

Round 1

Reviewer 1 Report

The manuscript presents an important theme that will contribute a lot to the specific application in nephrological patients, however, the following points must be observed:

1) Authors must follow the template and guide for publication;

2) the presentation of the results in the figures should be clearer and more explanatory;

3) Describe the research approval information by the ethics committee, for example, protocol number, etc...

4) it is essential to correlate the pharmacological mechanism of action of the anticoagulant with the pathophysiological aspects of nephropathy, in the way that only specialists can understand the importance of this therapeutic approach.

These observations are of great importance for the publication of the manuscript.

Author Response

The manuscript presents an important theme that will contribute a lot to the specific application in nephrological patients, however, the following points must be observed:

  • Authors must follow the template and guide for publication;

How to suggest the author’s guide I follow the below instructions for an article: Abstract, Keywords, Introduction, Materials and Methods, Results, Discussion, and Conclusions. I’ve reviewed the reference’s style and they are now in ACS style. Please let me know If you consider it necessary re-check something more.

2) the presentation of the results in the figures should be clearer and more explanatory;

I’ve modified both figures improving, at my notice, the quality of the legends and the explanatory.

3) Describe the research approval information by the ethics committee, for example, protocol number, etc...

I’ve added in the methods paragraph the below information:

"The study was approved, with the code 21/20, by the ethics committee for drug research of the Dr Peset University Hospital, Valencia on 03/30/2020. The same committee approved the exemption from requesting informed consent from patients due to the study’s retrospective nature. The main authors of the study conducted the statistical analysis. The first draft of the manuscript was written by the first and last authors and was revised by the co-authors. The decision to submit the manuscript for publication was made jointly by all the authors. The first and last authors voucher for the data's completeness and accuracy and for the protocol's fidelity to the trial."

4) it is essential to correlate the pharmacological mechanism of action of the anticoagulant with the pathophysiological aspects of nephropathy, in the way that only specialists can understand the importance of this therapeutic approach.

I’ve added in the introduction paragraph the below information:

"Despite the increase in the arsenal of anticoagulant drugs, the treatment of choice for patients with ACKD has not yet been established. There is a concern about increased vascular calcification and calciphylaxis with VKAs given that it reduces the function of vitamin K–dependent vascular calcification inhibitors and about the possibility of AKI secondary to glomerular haemorrhage due to VKAs and to a lesser extent to DOACs in whom there is no other identifiable aetiology of AKI. (6)

Furthermore, the pharmacokinetics and pharmacodynamics of most anticoagulant drugs are significantly affected by renal dysfunction, furthermore, they have significant drug-drug interactions that are especially important given the polypharmacy that is so prevalent in patients with CKD. (6)

These observations are of great importance for the publication of the manuscript.

Reviewer 2 Report

This manuscript describes a cohort study in which the authors enrolled 578 patients with both chronic kidney disease (CKD) and atrial fibrillation (AF) and they analyzed the clinical data, finding the treatment with direct-acting oral anticoagulants (factor Xa inhibitors and thrombin inhibitors) was associated with fewer ischemic-embolic complications than the treatment with warfarin. The idea of comparing the efficacy of direct-acting oral anticoagulants and warfarin on AF-related adverse events in CKD patients is pretty new. And the large sample size and multi-center design is decent, all of which brings this study some novelty and scientific significance. I listed several major concerns need to be addressed. 

1.    In the introduction part line 41, “There is a notable lack of published clinical trials in patients with CKD stages 4, 5 and 5D in treatment with DOACs”. Is this study specifically about the end stage of CKD? I didn’t see this specificity in the method part.

2.    Did the enrolled patients accept hemodialysis? What is the medication for the enrolled patients except for the anticoagulation therapy? More treatment details are needed in the method part.

3.    Please state the outcome/end-point of the retrospective follow-up and explain the meaning of “final” of the panel (basal/one year/final) in Table 1.

4.  For the figure 1 and figure 2, please translate the X and Y axis labels to English. And show the full name of FGe in figure 2.

Author Response

In the introduction part line 41, “There is a notable lack of published clinical trials in patients with CKD stages 4, 5 and 5D in treatment with DOACs”. Is this study specifically about the end stage of CKD? I didn’t see this specificity in the method part.

The study included all the patients followed up in the Nephrology department. According to the Spanish consensus document for the detection and management of chronic kidney disease, patients with eGFR < 30 ml/min/m2 or, among others, with higher eGFR but rapid progression (creatinine albumin ratio > 300 mg /g or decrease in eGFR > 5ml/min/year) are usually referred to the Nephrology department.

For this reason, most patients included in the study are CKD stages 3b and 4.

  1. Did the enrolled patients accept hemodialysis? What is the medication for the enrolled patients except for the anticoagulation therapy? More treatment details are needed in the method part.

We didn’t include in the study patients in haemodialysis. I copy the specificity of the method part about that: “Patients on renal replacement therapy, including kidney transplant, with valvular disease defined as moderate to severe mitral or aortic regurgitation, with stenosis, or prosthetic valve carriers were excluded from the study”. If you think it’s clearer I can add to the renal replacement therapy the specifications: haemodialysis and home dialysis

The use and the effect of anticoagulants in dialysis patients wasn’t the study’s goal; furthermore, in Spain, only Acenocoumarol is authorized for no valvular AF in haemodialysis and the use of DAOCs, in these patients, in 2020 was scarce.

For the rest of the patients, the primary objective of the study was the prevalence of AF in a nephrologist clinic and its complications, we didn’t have more details about the treatment except the anticoagulant medications.  

  1. Please state the outcome/end-point of the retrospective follow-up and explain the meaning of “final” of the panel (basal/one year/final) in Table 1.

I’ve specified the meaning of “final” (End of follow-up – the last visit at the moment recollecting the data). I’ve also modified the term in Figure 1

  1. For the figure 1 and figure 2, please translate the X and Y axis labels to English. And show the full name of FGe in figure 2.

Thanks, I’ve corrected Figures 1 and 2.

Reviewer 3 Report

The manuscript is acceptable for publication after a minor review:

1) Remove the word Introduction from the abstract section

2) Improve the resolution of Fig 1 and Fig 2

3) Change the referencing style

4) Improve the language and grammar

Author Response

  • Remove the word Introduction from the abstract section

Thanks. I’ve corrected that.

  • Improve the resolution of Fig 1 and Fig 2

I’ve modified both figures improving, at my notice, the quality of the legends and the explanatory

3) Change the referencing style

I've rewritten the references in ACS style

4) Improve the language and grammar

The manuscript was reviewed by a native and professional English translator. Anyway, I made an effort to review the text.

Round 2

Reviewer 1 Report

The authors complied with all the highlighted observations and the manuscript is in condition for publication.

Author Response

I'm really grateful with your revision and I thank you for the effort, time and energy, in reviewing the manuscript.

Reviewer 2 Report

This manuscript describes a cohort study in which the authors enrolled 578 patients with both chronic kidney disease (CKD) and atrial fibrillation (AF) and they analyzed the clinical data, finding the treatment with direct-acting oral anticoagulants (factor Xa inhibitors and thrombin inhibitors) was associated with fewer ischemic-embolic complications than the treatment with warfarin. The idea of comparing the efficacy of direct-acting oral anticoagulants and warfarin on AF-related adverse events in CKD patients is pretty new. And the large sample size and multicenter design is decent, all of which brings this study some novelty and scientific significance. The authors responded well to my questions and made some revisions. After explanations, the conclusion is less vulnerable. Overall, the manuscript offers abundant data and the tables and figures are logically organized. I listed several concerns need to be addressed. 

1.    Please clarify the total follow-up time for all the patients or the time-length for the observation in the method parts, in order to eliminate the leading-bias.

2.    Factor Xa and thrombin have been found to induce the inflammatory activity of macrophage in endothelium (PMID: 14736878). Macrophages play a central role in the atherosclerosis-related inflammation and trigger the development of coronary artery disease (PMID: 34197316, 33031913). This idea can help to explain why the Factor Xa inhibitors and thrombin inhibitors treatment was associated with fewer coronary events and long-term mortality. For the readers’ benefit and interest, it is suggested to discuss more about this in the discussion part.

Author Response

Please clarify the total follow-up time for all the patients or the time-length for the observation in the method parts, in order to eliminate the leading-bias.

In the results paragraph, complications during the follow-up, you can find the follow-up time for all patients: “Median follow-up from AF diagnosis to the most recent consultation was 61.0 months (33.0–101)”

Anyway, I’ve added a specification more about the time of the follow-up in the methods part:

The FAERC project was designed as a retrospective, multicentre, observational study of historical cohorts in which data was collected from arrhythmia diagnosis onwards (until the last visit consultation on June 30th 2020).

  1. Factor Xa and thrombin have been found to induce the inflammatory activity of macrophage in endothelium (PMID: 14736878). Macrophages play a central role in the atherosclerosis-related inflammation and trigger the development of coronary artery disease (PMID: 34197316, 33031913). This idea can help to explain why the Factor Xa inhibitors and thrombin inhibitors treatment was associated with fewer coronary events and long-term mortality. For the readers’ benefit and interest, it is suggested to discuss more about this in the discussion part.

I agree with your comment. If you consider it fine, I discuss the role of antivitamin-K in the progression of vascular calcification as one of the main risk factors of vascular events.

“However, our results go in the same direction as previously published studies in which the use of aVK (acenocoumarol and warfarin) is associated with an increase in systemic calcification, including in the coronary and peripheral vasculature. This increase in vascular calcification is due to the inhibition of the enzyme matrix Gla protein (MGP). MGP is a vitamin K-dependent protein that normally prevents systemic calcification by removing calcium phosphate from tissues. AVK induced systemic calcification would be related to an increase in vascular events (cardiac and peripheral).”

  1. Timothy J. Poterucha and Samuel Z. Goldhaber. 2015. Warfarin and vascular calcification. (December 2015). Retrieved November 26, 2022 from https://www.amjmed.com/article/S0002-9343(15)30031-0/fulltext